# The H163A mutation unravels an oxidized conformation of the SARS-CoV-2 main protease

Norman Tran[1,5], Sathish Dasari [2,5], Sarah A. E. Barwell[1], Matthew J. McLeod [3], Subha Kalyaanamoorthy[2,6], Todd Holyoak [1,6] ✉ & Aravindhan Ganesan [4,6] ✉

The main protease of SARS-CoV-2 (Mpro) is an important target for developing COVID-19 therapeutics. Recent work has highlighted Mpro's susceptibility to undergo redox-associated conformational changes in response to cellular and immune-system-induced oxidation. Despite structural evidence indicating large-scale rearrangements upon oxidation, the mechanisms of conformational change and its functional consequences are poorly understood. Here, we present the crystal structure of an Mpro point mutant (H163A) that shows an oxidized conformation with the catalytic cysteine in a disulfide bond. We hypothesize that Mpro adopts this conformation under oxidative stress to protect against over-oxidation. Our metadynamics simulations illustrate a potential mechanism by which H163 modulates this transition and suggest that this equilibrium exists in the wild type enzyme. We show that other point mutations also significantly shift the equilibrium towards this state by altering conformational free energies. Unique avenues of SARS-CoV-2 research can be explored by understanding how H163 modulates this equilibrium.

Severe acute respiratory syndrome coronavirus 2 (SARS-CoV-2) is the etiological agent of COVID-19[1], a disease that has resulted in at least 500 million cases and six million deaths worldwide as of June 2022[2]. In wake of this global pandemic, the structural biology scientific community has pivoted their research towards discovering and developing therapeutics[3] and vaccines[4] against target proteins essential for SARS-CoV-2 replication[5]. These endeavors rely on a fundamental understanding of the structure–function relationships of many of these viral proteins, which, especially in light of the ever-increasing diversity of new SARS-CoV-2 variants[6], must be continuously enriched to aid in the discovery and design of more efficacious therapeutics[7].

The main protease (Mpro; also known as Nsp5) of SARS-CoV-2 is one such protein essential to SARS-CoV-2 replication[8]. Mpro is an obligate homodimeric cysteine protease which functions to cleave the viral polyproteins into their individual functional components necessary for viral replication[9]. Because of its critical role in polyprotein processing, the virus is unable to replicate when the enzyme is inhibited[10]. Furthermore, Mpro is highly conserved amongst the known coronavirus family and shares no structural homology or substrate specificity with the human proteome[11]. Mpro, therefore, remains a very attractive target for therapeutic research.

Structurally, each Mpro protomer can be divided into three domains (Fig. 1a): Domain I (residues 1–101), Domain II (102–184), and Domain III (201–301)[12]. The active site is located at the interface between Domain I and Domain II, where H41 in Domain I and C145 in Domain II make up the catalytic dyad (Fig. 1b)[13]. Mpro is an obligate dimer because the N-terminus (also called the N-terminal finger) of the second protomer closes the active site of the first protomer.

[1]Department of Biology, Faculty of Science, University of Waterloo, 200 University Avenue West, Waterloo, ON N2L 3G1, Canada. [2]Department of Chemistry, Faculty of Science, University of Waterloo, 200 University Avenue West, Waterloo, ON N2L 3G1, Canada. [3]Physics Department, Cornell University, Ithaca, NY 14853, USA. [4]ArGan's Lab, School of Pharmacy, Faculty of Science, University of Waterloo, 10A Victoria Street South, Kitchener, ON N2G 1C5, Canada. [5]These authors contributed equally: Norman Tran, Sathish Dasari. [6]These authors jointly supervised this work: Subha Kalyaanamoorthy, Todd Holyoak, Aravindhan Ganesan. ✉e-mail: tholyoak@uwaterloo.ca; aravindhan.ganesan@uwaterloo.ca

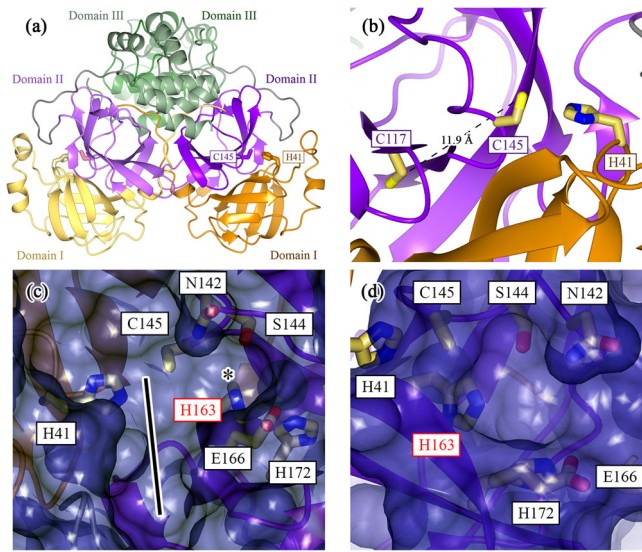

**Fig. 1 | Structure of the SARS-CoV-2 Mpro highlights the importance of the lateral pocket in inhibitor design.** Mpro is an obligate homodimeric cysteine protease (PDB 7BB2). **a** Each monomer can be broken up into three regions: Domain I (residues 1–101; yellow/orange); Domain II (102–184; light violet/magenta); and Domain III (201–301; pale green/forest green). **b** The active site in each monomer is created from the interface between Domains I and II, whereby the catalytic dyad's H41 and C145 are derived from Domains I and II, respectively. In the WT structure, the active-site cysteine (C145) is located ~12 Å from C117, the cysteine involved in the disulfide bond in the H163A Mpro structure. **c** Surface representation of the WT Mpro with a focus on the active-site cleft. The enzyme's S2 to S4 pockets are denoted by the black line. The key residue of interest, H163, is located in the S1 pocket, laterally connected to this active site groove (denoted by *). **d** The surface representation from (**c**) is rotated 90° counterclockwise to show this H163 lateral pocket from a head-on perspective. Side chains that make up the lateral pocket and the catalytic dyad are rendered as cylinders in both (**c**) and (**d**). All molecular representations in this paper were generated in CCP4MG (version 2.10.11)[77].

Therefore, the enzyme exhibits varying levels of positive kinetic cooperativity depending on the specific substrate[14]. This highlights the importance of dimerization for Mpro enzymatic activity and the asymmetric communication between the two protomers, which manifests structurally and kinetically[15].

Mpro normally resides in the cytoplasm, a generally reducing environment where the reduction-oxidation (redox) potential is tightly controlled by the ratio of reduced to oxidized glutathione[16]. Reactive cysteine side chains, like Mpro's active-site nucleophile, C145, can undergo several consecutive reactions with molecular oxygen, reversibly transforming from a free sulfhydryl first to mono-oxygenated sulfenic acid, and upon over-oxidation, irreversibly transforming into di-oxygenated sulfinic and tri-oxygenated sulfonic acid[17]. The native cytoplasmic location and its corresponding reducing environment ensure that Mpro's cysteine side chains remain in their sulfhydryl form, allowing the active-site nucleophile to remain catalytically active[18]. More recently, direct crystallographic and indirect mass-spectrometric structural data have uncovered Mpro's structural and functional sensitivity to its redox environment[18–21]. This factor has often been overlooked in Mpro despite the role of redox in the regulation, allostery, and dimerization of other proteins[22–24]. Despite Mpro normally residing in a reducing environment, Mpro can be acutely exposed to bursts of reactive oxygen species and may be irreversibly inactivated via over-oxidation as a result[18]. These acute oxidative bursts are seen during heavy cellular respiration or as a defensive response from the innate immune system[25,26].

A few key redox-sensitive residues have been identified with direct crystallographic and indirect structural evidence. The active-site nucleophile has been shown to form peroxysulfenic acid (R-S-OOH)

when reducing agent is removed after the crystal has formed (PDB 6XB0)[18]. Mpro's C-terminal C300 is also susceptible to post-translational glutathionylation, seen alongside a minor proportion of glutathionylated C85 and C156 in mass spectrometry experiments, which hinder dimerization and activity[21]. Nitrogen-oxygen-sulfur (NOS; PDB 6XMK) and sulfur-oxygen-nitrogen-oxygen-sulfur (SONOS; PDB 7JR4) linkages have been identified in Mpro, connecting the side chains of residues K61/C22 and C44/K61/C22 with bridging oxygen atoms in NOS and SONOS bridges, respectively[19,20]. These linkages were described retrospectively by analyzing previously published Mpro structures and may play a role in protecting the enzyme against further irreversible oxidative damage[20,27].

Indirect mass-spectrometric results from Funk et al. highlight the possibility of even larger conformational changes associated with oxidation[20]. Their mass-spectrometric results show indirect structural evidence for the formation of an intramolecular disulfide bond between the active-site cysteine, C145, and a distant cysteine, C117, when the enzyme is treated with exogenous oxidizing agents[20]. C117 is not in close proximity to C145 (Fig. 1b) as their backbones are bridged by hydrogen bond interactions with the side chain of N28, which obstruct any direct contact between the two cysteines. However, an earlier study investigating the effect of the N28A mutation in the SARS-CoV-1 Mpro detailed a crystal structure (PDB 3FZD) depicting a disulfide bond between C117 and C145 and a collapsed oxyanion hole (involving residues G143 and S144)[28]. The dimerization constant of the N28A mutant was also increased ~19-fold compared to the wild-type (WT) enzyme[28]. Although the aforementioned indirect structural evidence suggests the existence of a similar oxidized state in SARS-CoV-2, this disulfide-bonded, oxidized structure has yet to be revealed and the underlying mechanism by which this conformational change occurs also remains unclear.

Due to the structural similarities between Mpro and homologous enzymes found in other coronaviruses, a vast number of putative drug compounds and fragments against the SARS-CoV-2 Mpro have already been structurally and biochemically characterized[29,30]. This has been achieved through the experimental and computational efforts of many groups, who have provided a wealth of information about specific pockets and residues that can be readily exploited in structure-based drug design approaches[31,32]. Our previous computational work has identified several key residues that greatly contribute to the binding energies of many well-characterized small molecules[33]. Many of these important residues are found in a pocket laterally connected to the active site (Fig. 1c, d)[33,34]. The H163 side chain was found to be a particularly important contributor to inhibitor binding as mutating it to an alanine (H163A) showed notable decreases in in silico binding affinity for five known inhibitors ranging from 1 to 9 kcal/mol[33]. These in silico results suggest that this lateral pocket can be exploited in rational structure-based drug design strategies by extending known inhibitors into this pocket to engage the site using chemical moieties that are known to interact strongly with H163.

This work aimed to biochemically characterize the effects of the H163A mutation on the structure and ligand binding capabilities of this mutant in vitro. Our preliminary findings on the H163A mutant show significant biophysical and kinetic differences relative to the WT enzyme despite having a near-identical structure when co-crystallized with the covalent inhibitor GC376. The structure of the H163A mutant reveals large-scale conformational changes and several oxidized side chains alongside a disulfide bond between C117 and the active-site nucleophile, C145. This oxidized structure is seen despite the mutant being purified, stored, and crystallized in a reducing agent (0.5 to 1.0 mM TCEP). Although a disulfide-bonded, oxidized conformation was previously reported in a point mutant of the SARS-CoV-1 Mpro[28], this work demonstrates that this oxidized conformation can also be observed by mutating a residue (H163) not directly in contact with the active-site residues. Using a combined analysis of structural data and metadynamics simulations, we propose a working hypothesis for the

mechanism by which this oxidized conformation occurs in the H163A mutant. This reiterates the importance of H163 in shaping the catalytic site and, henceforth, influencing the activity of the SARS-CoV-2 Mpro. We have evidence to suggest that this conformation can also occur in the WT enzyme, albeit at a much lower proportion compared to the H163A mutant, indicating that this mutant structure can potentially be used as a conformational target in structure-based drug design strategies.

## Results and discussion

### Structure of H163A Mpro in complex with GC376

To test the hypothesized role of H163 in contributing to inhibitor binding affinity, we first co-crystallized and solved the structure of the H163A mutant with GC376 (PDB 8DD6; Supplementary Table 1), a covalent inhibitor with an $IC_{50}$ value of $0.20 \pm 0.04\,\mu M$ against the WT enzyme[14]. The structure of the H163A mutant in the complex with GC376 is nearly identical to the WT GC376 complex (PDB 7TGR; Fig. 2a), with a Cα-RMSD of 0.38 Å across all residues. Cα RMSD values were calculated using Chimera (version 1.16–42360)[35]. The pose of the inhibitor is also nearly identical (Fig. 2b), retaining many of the same inhibitor-enzyme interactions as the WT complex (Supplementary Fig. 1). There are some minor rotameric and ring-puckering differences in some of the inhibitor moieties, likely due to differences in these interactions and potentially the inhibitor's local microenvironment between the two structures. The phenyl moiety in GC376 for the H163A mutant structure distinctly occupancies two conformations, while only one conformation is seen in the WT complex. This is not surprising given there are no enzyme-inhibitor interactions that directly contact this ring in either structure (Supplementary Fig. 1). A sole water molecule compensates for the loss of the H163 side chain by sterically occupying the now-empty side chain pocket while making hydrogen bond contacts with the inhibitor and protein backbone (Fig. 2c). Overall, these structures show that GC376 binds with the same orientation in both structures and minor adjustments to the enzyme are made in the H163A mutant due to the absence of the H163 side chain.

### Kinetic and biophysical characterization of H163A Mpro

Despite the kinetic characterization of the WT Mpro yielding kinetic and cooperativity constants agreeable with literature values (Supplementary Fig. 2)[14], the H163A mutant showed no detectable activity when assayed under similar conditions. The mutant was also inactive at this enzyme concentration when assayed with 10 mM β-mercaptoethanol, DTT, or TCEP after 1-h incubation at room temperature. Enzymatic activity was only seen when the mutant enzyme concentration was increased significantly, leading to a calculated $k_{cat}$ of ~30 times lower than WT. This was initially surprising given that the structure of the mutant complex with GC376 showed a near-identical conformation to the WT complex (Fig. 2 and Supplementary Fig. 1). However, this loss of catalytic activity is comparable with the in silico shift in equilibrium between active and inactive states induced by the H163A mutation, as described below.

Based upon these observed kinetic differences, differential scanning calorimetry (DSC) was used to quantitatively compare the thermal stability of the WT and H163A Mpro. The analysis demonstrated a clear shift in the temperature associated with the thermogram peak between the WT (57.2 °C) and H163A Mpro (54.0 °C) (Supplementary Fig. 3). A similar trend was seen with the DSC melting temperature of the N28A SARS-CoV-1 Mpro, whereby the point mutant showed a decrease by 1.8 °C relative to the WT SARS-CoV-1 protease[28]. The asymmetry and broadening of both thermogram peaks can be a result of a concerted, cooperative homo-oligomeric unfolding process, which has been shown to be a good model to fit Mpro thermograms[36]. Unfortunately, because the thermal denaturation process of the WT and H163A enzyme were irreversible, thermodynamic information (e.g., the melting temperature, enthalpy, and entropy of unfolding) and general mechanisms of unfolding could not be derived from these thermograms[28,37]. Despite this, these data show a clear difference in thermal stability between the two enzymes and suggest that the H163A mutant is well-folded and dimeric despite its relative kinetic inactivity.

To further verify the oligomeric state of the H163A mutant, small-angle X-ray scattering (SAXS) experiments were performed at concentrations ranging from 0.25 to 6.30 mg/mL. These SAXS profiles (Supplementary Fig. 4 and Supplementary Table 2) show that the mutant is indeed dimeric, consistent with a previously published WT Mpro SAXS dataset (SASBDB Entry SASDJG5)[38,39], and was notably different from the monomer generated by removing one protomer from the known dimer structure (Supplementary Fig. 5). This suggests that the monomer-dimer equilibrium is not significantly affected by the H163A mutation at the concentrations of enzyme used in the DSC and SAXS experiments.

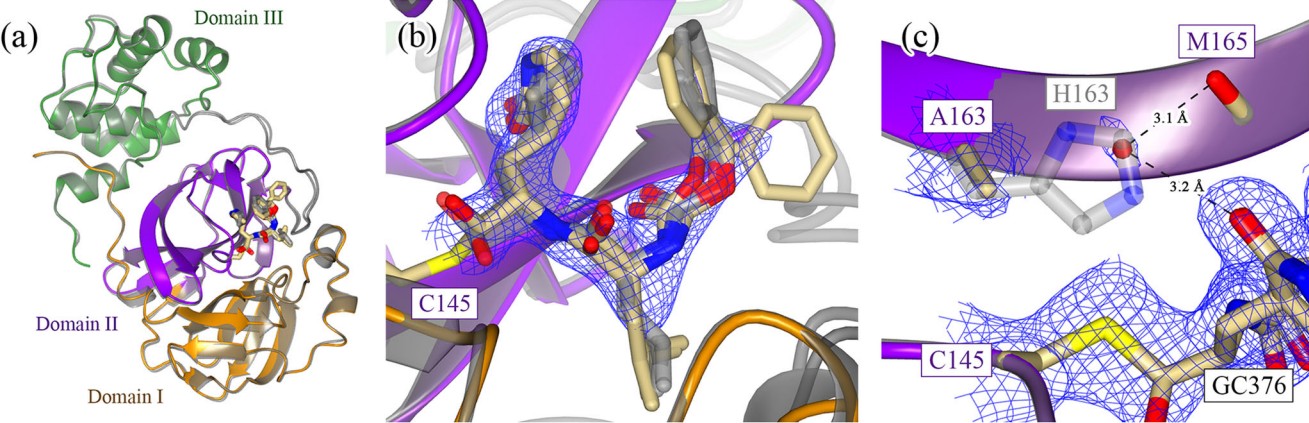

**Fig. 2 | Structure of wildtype and H163A Mpro in complex with GC376. a** The general fold of Mpro is conserved when comparing the WT (gray; PDB 7TGR) and H163A mutant structures in complex with the covalent inhibitor GC376 (PDB 8DD6). Only one monomer is depicted as the other monomer comprising the dimer is crystallographically identical. **b** The pose of GC376 is also nearly identical between the WT (gray) and H163A mutant structures, although there are slight differences in the ring puckering and rotameric conformation of some inhibitor moieties, particularly the phenyl ring of GC376. These changes are supported by 2$F_o$-$F_c$ density at 1.2 σ and can be attributed to the slightly different inhibitor-enzyme interactions between the two structures (Supplementary Fig. 1). **c** The carbonyl group of the γ-lactam moiety makes a hydrogen bond with the H163 imidazole ring in the WT enzyme (gray). Upon mutation to alanine, a water molecule compensates for the loss of the imidazole ring by making hydrogen bonds with GC376 and the backbone carbonyl of M165. This is supported by 2$F_o$-$F_c$ density at 1.0 σ.

## Crystal structure of H163A Mpro shows an inactive oxidized conformation

As there are clear differences in the kinetic activity and thermal stability between the WT and H163A mutant, we determined the crystal structure of the H163A mutant (PDB 8DDL; Supplementary Table 1) to investigate structure differences that could account for these biophysical changes. The mutant enzyme takes on the same global three-domain fold as the WT structure (PDB 7BB2; Supplementary Fig. 6a), with a Cα-RMSD of 1.13 Å across all residues. Despite this, the H163A mutation triggered large and important structural changes in the catalytic center and overall repositioning of Domain III relative to Domains I and II (Supplementary Fig. 6b). Notably, the H163A mutant shows three main structural differences that are not present in the WT enzyme: (1) a disulfide bond involving the active-site nucleophilic cysteine and the rearrangement of nearby residues, (2) the formation of a NOS bridge distal to the active site, and (3) the rethreading of four amino acids at the N-terminus of each protomer.

(1) The most interesting of these structural differences occur in the loops of Domain II, where the active site and the surrounding residues are restructured to accommodate a newly formed disulfide bond between the active-site nucleophile, C145, and the now-adjacent C117 residue (Fig. 3a), as observed in the crystal structure of N28A SARS-CoV-1 Mpro[28]. The side chain of N28 changes its rotameric state so that it does not sterically hinder the disulfide bond (Fig. 3a). In the WT enzyme, N28 interacts with the backbone carbonyl of both C145 and C117, a network that was shown to be important in the dimerization of Mpro[28]. While this disulfide bond has been observed in the structure of the N28A SARS-CoV-1 Mpro[28], our structure demonstrates that a similar conformation can be triggered by mutating a residue in the lateral pocket, whose side chain does not make close contacts with C117 or C145. The F140 residue, whose side chain normally faces inwards to form a π-stacking interaction with H163 in the WT Mpro, is significantly displaced and its side chain becomes surface exposed in the mutant structure. This dislocates the oxyanion loop located N-terminal to the active-site cysteine, C145. Neither N28 or C117 are fully occupied in their rotated conformation or in a disulfide bond (Fig. 3a), respectively. There is also structural evidence that shows that, upon breakage of the disulfide bond, the β-strand containing the active-site cysteine (C145) relaxes to a more WT-like conformation as there is positive F$_o$-F$_c$ density to support the strand in the relaxed conformation (Fig. 3b, c). Unfortunately, there was additional positive F$_o$-F$_c$ density even when this alternate conformation was modeled and refined, likely due to the partial occupancy of solvent molecules in the same pocket when the disulfide bond is formed. Due to our inability to model this alternate conformation accurately, no atoms were placed in this positive F$_o$-F$_c$ density. C117 similarly changes its rotameric state upon breakage of the disulfide bond (Fig. 3a). Despite there being positive F$_o$-F$_c$ density adjacent to the sulfur atom of the free C117, it is unclear whether this density can be attributed to sulfenic acid or a small proportion of the beta strand containing C117 in a WT-like, relaxed conformation (Supplementary Fig. 7). Regardless, these events seem to be restricted to their local environment as no other part of the mutant structure shows evidence of partial occupancy.

The fact that the mutant structure shows a mixed population of broken and intact disulfide bond suggests that the formation of the disulfide bond may be reversible under reducing conditions. However, given the kinetic inactivity of the mutant relative to the WT enzyme, the disulfide-bonded, oxidized conformation seems to be the dominant species in solution at equilibrium. The existence of this reversible reaction explains why the H163A mutant was able to co-crystallize with GC376. As the small proportion of mutant enzyme with a free active-site sulfhydryl can react with GC376 irreversibly, all mutant Mpro molecules are eventually sequestered into a complex with GC376 despite the apo mutant primarily being in the disulfide-bonded conformation. The stabilizing interactions that GC376 has across the entire enzyme (Fig. 2b and Supplementary Fig. 1), most notably through GC376's ability to covalently link to C145, stabilize the oxyanion hole via G143, and occupy the lateral pocket, allows the mutant to regain a WT-like conformation[14,40]. However, the question of if the binding of a non-covalent ligand that does not engage with the lateral pocket will enable Mpro to regain a WT-like state still remains unanswered.

(2) The mutant structure also shows several interesting features distal to the active site. A NOS linkage is seen in one of the two chains in the mutant structure (Fig. 4a, b). This has been seen indirectly through mass spectrometry and directly in crystallo for the SARS-CoV-2 Mpro[19,20] but not for the N28A SARS-CoV-1 Mpro, despite the disulfide bond[28]. The role of NOS and SONOS

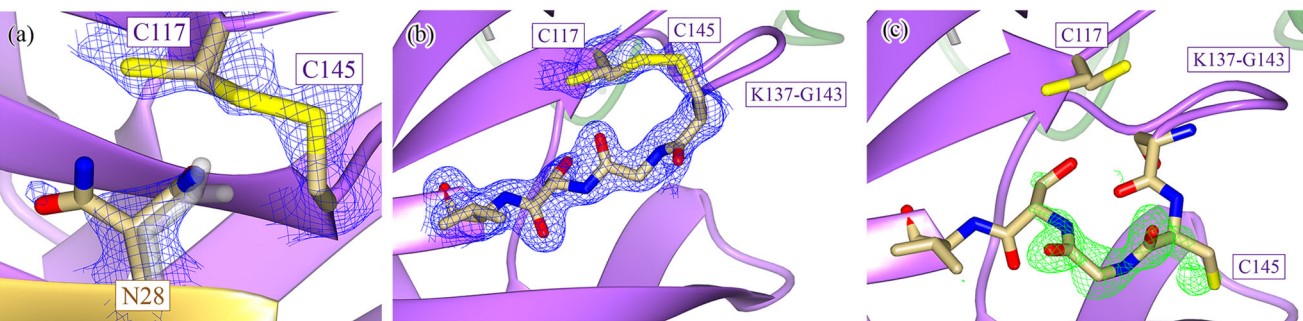

**Fig. 3 | The active-site nucleophile, C145, is protected in a disulfide bond in the H163A Mpro structure. a** The H163A Mpro structure contains a disulfide bond between the active-site nucleophile, C145, and the previously distant cysteine C117 (Fig. 1b). This disulfide bond is not completely formed as there is 2F$_o$-F$_c$ density at 1.2 σ that supports an alternate, non-disulfide-bonded conformation of C117. The side chain of N28 also takes on two conformations, one seen in both WT (gray) and mutant structures and the other only seen in the mutant. There is a concomitant structural change between the formation of the disulfide bond and the rotation of the N28 side chain, as the N28 side chain in its WT conformation sterically hinders the disulfide bond from forming. The beta strand containing the active-site nucleophile can take on two conformations depending on whether or not the disulfide bond is present. **b** When the disulfide bond between C145 and C117 is formed, the C145 beta strand runs antiparallel to the C117 beta strand (2F$_o$-F$_c$ density shown at 1.2 σ). **c** When the disulfide bond is broken, the C145 beta strand relaxes to a second conformation (F$_o$-F$_c$ density shown at 4.0 σ), aligning almost exactly to the WT conformation of the strand when the structures are superimposed. The flexibility in the loop N-terminal to the C145 beta strand (K137-G143) allows for this relaxation to occur. Despite there being crystallographic evidence for both conformational states, only the disulfide-bonded conformation was modeled, as the density corresponding to the second conformation could not be accurately modeled with only one alternate conformation. **b, c** depict the two conformations of chain B of the H163A structure.

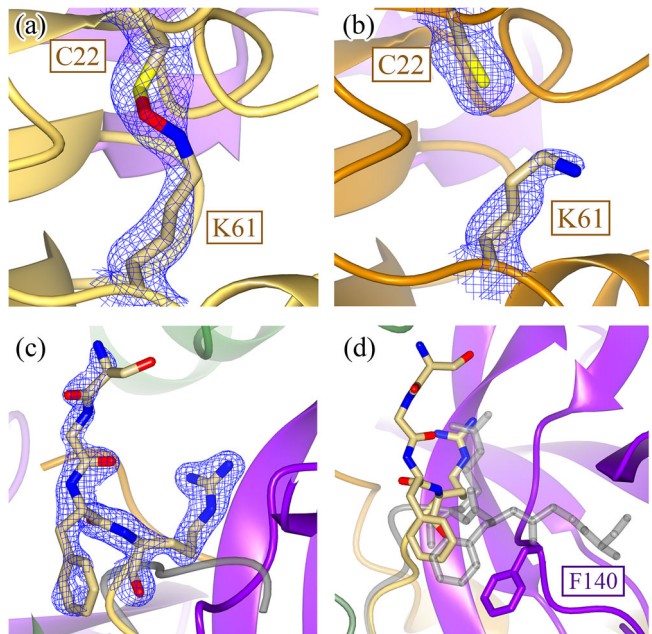

**Fig. 4 | Structural comparisons between wildtype and H163A Mpro structures distal to the active site.** In addition to the local restructuring of the active site, structural changes are also seen distally in Domain I. A NOS bridge between C22 and K61 is captured in chain B (**a**) ($2F_o$-$F_c$ density shown at 1.2 σ) but not in chain A (**b**) ($2F_o$-$F_c$ density shown at 1.0 σ). **c** This structural asymmetry is also seen when comparing the N-termini of the two monomers, where $2F_o$-$F_c$ density is only seen for the N-terminus of chain B (shown at 1.4 σ) but not chain A. **d** When comparing the positions of the N-termini between the WT (gray) and H163A mutant, the four most N-terminal residues are drastically rotated approximately 90° to fit into an alternate pocket. This is due to the movement of the F140 loop in the H163A mutant structure, which occupies the space previously held by the WT N-terminus. There is no density to support a single conformation of the N-terminus in the other protomer.

bridges can vary from being a conformational redox switch to an oxygen sensor for oxygen-sensitive proteins[19,27]. They can also act to stabilize domains, much like the role of most disulfide bridges, or provide reactive centers to protect against over-oxidation, both of which are possible functions of the NOS in Mpro[19,20,27,41].

(3) In contrast to the minor structural changes associated with NOS bridge formation, the N-terminus of one of the H163A mutant protomers undergoes a drastic conformational change whereby the backbone of the first four N-terminal residues thread through a completely different path, in a direction almost 90° relative to its WT conformation (Fig. 4c). This is a result of the structural repositioning of the active-site loop (S139-S147) and, in particular, the F140 side chain into the same position previously occupied by the WT N-terminus. This is not seen in the disulfide-bonded N28A SARS-CoV-1 Mpro structure, potentially due to the disorder seen in the F140 loop[28].

Many of the local conformational changes, like the disulfide bond and rearrangement of the N-terminus, can be directly attributed to the movement of the F140 side chain, a residue that is situated on the active-site loop (S139-S147). There is a large 14 Å motion of the F140 side chain from an inward (Fig. 5a, gray) conformation, as seen in the WT structure, to an outward (Fig. 5a, purple) conformation, as seen in the H163A mutant structure. This moves the F140 side chain from the core of the enzyme (inward conformation) to the outside of the enzyme, where it is solvent exposed and adjacent to the C-terminus of the other protomer (outward conformation). When the active-site loop

adopts the outward conformation, the now-open space previously occupied by the F140 side chain is replaced by a new hydrogen-bonding network formed from two water molecules and a few repositioned side chains (Fig. 5b).

Normally, this inward conformation is stabilized by a face-to-face π-stacking interaction between F140 and H163 (Fig. 5c). Our hypothesis is that the missing π-stacking interaction in the H163A mutant desta-bilizes the inward conformational state of F140, perturbing the ground-state conformational equilibrium such that it becomes more favorable to access the lower-energy outward conformation. When the F140 and the active-site loop are flipped outward, the mutant enzyme can stabilize this conformation by forming a disulfide bond with the active-site nucleophile, C145, as it is now placed adjacent to C117. This disulfide-bonded, oxidized conformation is what is seen in the apo H163A mutant structure.

When the mutant crystals were soaked with 20 mM TCEP for two hours, the resultant crystal structure (PDB 8SG6; Supplementary Table 1) was fully reduced as both the NOS bridge and the C117-C145 disulfide bonds were broken in both protomers (Supplementary Fig. 8a). Interestingly, despite the breakage of the disulfide bond, the active-site loop was not repositioned as the F140 side chain remained flipped in the outward position (Supplementary Fig. 8b). This suggests that the disulfide bond stabilizes this inactive state but is not sufficient for generating this conformation. The disulfide bond is thus a con-sequence of placing the two cysteine residues in proximity following the displacement of the active-site loop. Our work, therefore, indicates that the loss of the π-stacking interaction between F140 and H163 is the source of the inactive conformation of Mpro.

To summarize, this working hypothesis is supported by four experimental results. (1) The presence of crystallographic densities for both intact and broken disulfide bonds suggest that the disulfide bond can be reversibly formed under reducing conditions (Fig. 3a). (2) Kinetic differences between the WT and mutant enzyme show that the mutant exists primarily in an inactive disulfide-bonded state (Supplementary Fig. 2). In other words, the equilibrium for the mutant is shifted far towards the disulfide-bonded, oxidized conformation. (3) The ability of GC376 to regenerate a WT conformation in the GC376-bound mutant structure (Fig. 2a) confirms that the equilibrium can be pushed back towards a competent conformation by using an irreversible reaction, even when the active state of the mutant enzyme is found in a small proportion. And (4), the reduced mutant structure shows that the disulfide bond between C145 and C117 helps to stabilize the inactive conformation when F140 is in the outward position (Supplementary Fig. 8).

## Metadynamics reveals that the H163A mutation lowers the free-energy barrier for sampling the disulfide-bonded, oxidized conformation

We performed metadynamics simulations to understand the free-energy landscape and underlying mechanistic processes leading from the WT Mpro to the alternate conformational state. When comparing the apo WT and H163A mutant crystal structures, three important structural changes are observed: (1) exposure of the F140 side chain to the surface and dislocation of the active site loop (Fig. 5a); (2) dihedral rotation of the N28 side chain (Fig. 3a); and (3) C117-C145 disulfide bond formation. Since bond formation and breakage cannot be observed using classical molecular dynamics (MD) simulations, we used the first two structural changes as collective variables (CVs) to understand the conformational transitions observed.

In the WT structure (state "A"), the H163-F140 Cα distance is ~8 Å to allow their side chains to stack against each other and the side chain dihedral angle (C-Cα-Cβ-Cγ) of N28 is ~178° so as to make hydrogen bonds with the backbone carbonyls of C117 and C145. On the contrary, in the alternate conformation sampled by the H163A crystal structure (state "B"), the side chain of F140 is solvent exposed and the Cα distance

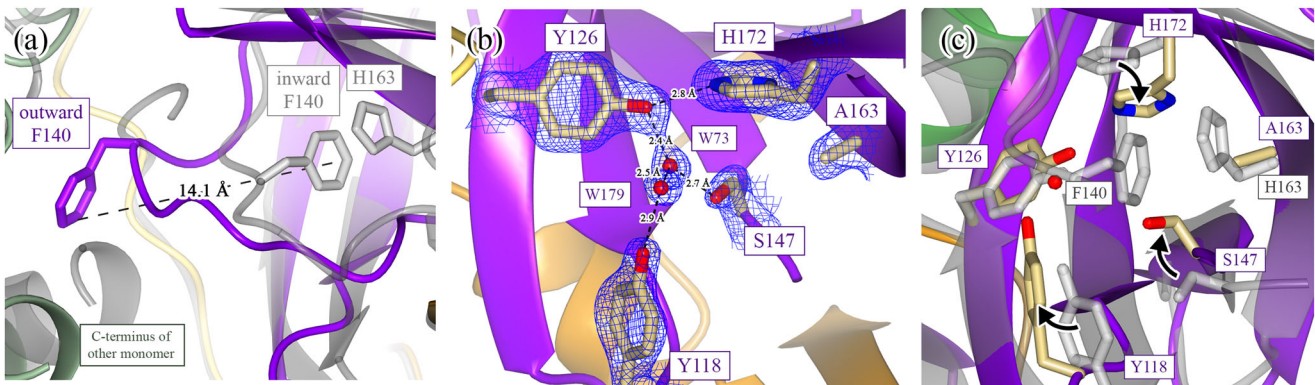

**Fig. 5 | Conformational changes of the F140 loop result in large-scale structural rearrangements.** Many of the structural differences seen between the WT and H163A Mpro structures can be attributed to the movement of the F140 side chain. **a** F140 is normally found in an inward conformation within the core of the enzyme (WT structure is in gray), where it is stabilized by a face-to-face π-stacking interaction with the side chain of H163. When the H163 side chain is mutated, F140 flips to an energetically favored outward conformation in an ~14 Å motion to situate itself close to the C-terminus of the other monomer. **b** The outward conformation in the H163A mutant results in the formation of a new hydrogen-bonding network in the space previously occupied by the F140 side chain (2F$_o$-F$_c$ density shown at 1.3 σ). This network is formed from two new water molecules (W73 and W179) and the side chains of Y118, Y126, S147, and H172. **c** An overview of the structural rearrangement of this F140 pocket is shown with arrows indicating the motion of these side chains from their starting (WT; gray) to ending (H163A) conformations.

between A163 and F140 elongates to ~14 Å (Fig. 5a); whereas the side chain dihedral angle of N28 is 60° as it clears the path for the disulfide bond between C145 and C117 (Fig. 3a). Therefore, starting from the WT structure, we performed our simulation to accelerate the separation of the Cα distance between H163 and F140 and the rotation of the N28 side chain using history-dependent bias potentials, as described in the methods. This helped us construct the free-energy profile and identify the minimum free-energy path from states "A" to "B". The 2D free-energy surface for the WT metadynamics simulation (Fig. 6a) reveals that state A (WT) corresponds to the global minimum, while state B (mutant-like conformation in WT) is located at a shallow local minimum. The 1D profile of the minimum path (Fig. 6b) extends insights into the free-energy barriers along this path. Initially, the dihedral rotation of N28, breaking its side chain interactions with C117 and C145, occurs with an energetic barrier of ~6 kcal/mol (Supplementary Fig. 9). This led the structure to a new minimum-energy state, in which the backbone of N28 only interacted with C145; however, this interaction broke as the structure passed through the second barrier corresponding to perturbation of the aromatic stacking of H163 and F140, and subsequent active-site loop dislocation triggered by H163-F140 separation (Supplementary Fig. 9). In summary, the state B is ~8 kcal/mol higher than the state A, indicating the unfavorable nature of the later conformation in the WT Mpro structure. This explains the absence of a crystal structure of a C117-C145 disulfide-bonded conformation of WT Mpro despite indirect evidence for the existence of such a conformation under oxidizing conditions, as described previously[20].

Next, we mutated H163 to alanine in the background of the WT Mpro structure and repeated the metadynamics simulation for this mutant model by using the same CVs to observe the transition from states "A" to "B". Interestingly, unlike in the WT Mpro, we observe the opposite trend for both the states in this mutant model (Fig. 6c). The WT-like conformation (state A) has higher free energy than the mutant conformation (state B). In fact, the 1D free-energy profile of the minimum-energy path for state transitions in the mutant model (Fig. 6b) shows that state B, which is close to the conformation from the H163A crystal structure (Fig. 6d), is ~8 kcal/mol lower than that of state A. During the metadynamics simulation of the H163A mutant model, the elongation of the A163-F140 Cα distance is the first step in this transition. This separation quickly occurred as it had no significant energetic barrier, which contrasted with a larger barrier (~6 kcal/mol) observed for the same process in the WT simulation. This is postulated to be due to the lack of π-π interactions between A163 and F140 that

was observed in the WT system. The H163A model reached a unique global minimum state B', in which the C117-N28-C145 interactions were intact, but F140 was exposed to the surface, and the active-site loop was displaced (Supplementary Fig. 10). Consequently, the main rate-limiting step (with a barrier of ~7 kcal/mol) for the conformational change in the mutant model corresponds to the side chain rotation of N28. Following this rotation, the simulation reaches state B, which is the second global minimum in this free energy path. We further noted that the distance between the sulfur atoms of C117 and C145 fell from 11 to 9 Å during the state transitions in the mutant model (Supplementary Fig. 10). This suggests that in the absence of the H163-F140 aromatic stacking in the mutant model, the barrier for displacing the active-site loop is lowered. Displacement of F140 into the outward conformation allows the catalytic cysteine, C145, to disulfide bond with C117, leading to what is observed in our crystal structure.

It should be noted that F140 has been previously seen in a partially exposed conformation for an immature form of Mpro which included three extra uncleavable N-terminal residues to simulate an intermediate state along the Mpro maturation process[42,43]. Despite this partial exposure of F140 (H163-F140 Cα distance = 10.5 Å), no structural rearrangements related to oxidation of C145/C117 were seen[43,44]. This suggests that there is more nuance in understanding the origin and functional consequences of the conformational changes associated with F140 and the active-site loop. We, therefore, tested the effects of breaking the H163-F140 aromatic stacking via mutation of F140 on the conformational transition of Mpro using metadynamics. For this purpose, we built an F140A mutant model from our WT structure and subjected it to metadynamics with the same set of CVs as described above. The free-energy landscape corresponding to the A→ B state transitions in the F140A model was also altered (Fig. 6b), with the major barrier of ~7 kcal/mol corresponding to the rotation of the N28 side chain (Fig. 6e and Supplementary Fig. 11). However, the question of if the F140A point mutant will also follow the same mechanism to lead to a similar oxidized conformational state is yet to be addressed.

Finally, we tested the effects of N28A mutation while retaining the H163-F140 aromatic stacking intact within the lateral pocket. Understandably, we only involved one CV pertaining to the elongation of H163-F140 Cα distance as N28 was mutated to an alanine. The free-energy landscape for the N28A model (Fig. 6f) showed a quick transition from state A to state B, with state B being ~5 kcal/mol lower in energy (closer to the free-energy differences between states A and B in WT and other mutant models, Fig. 6b). As such, there was no

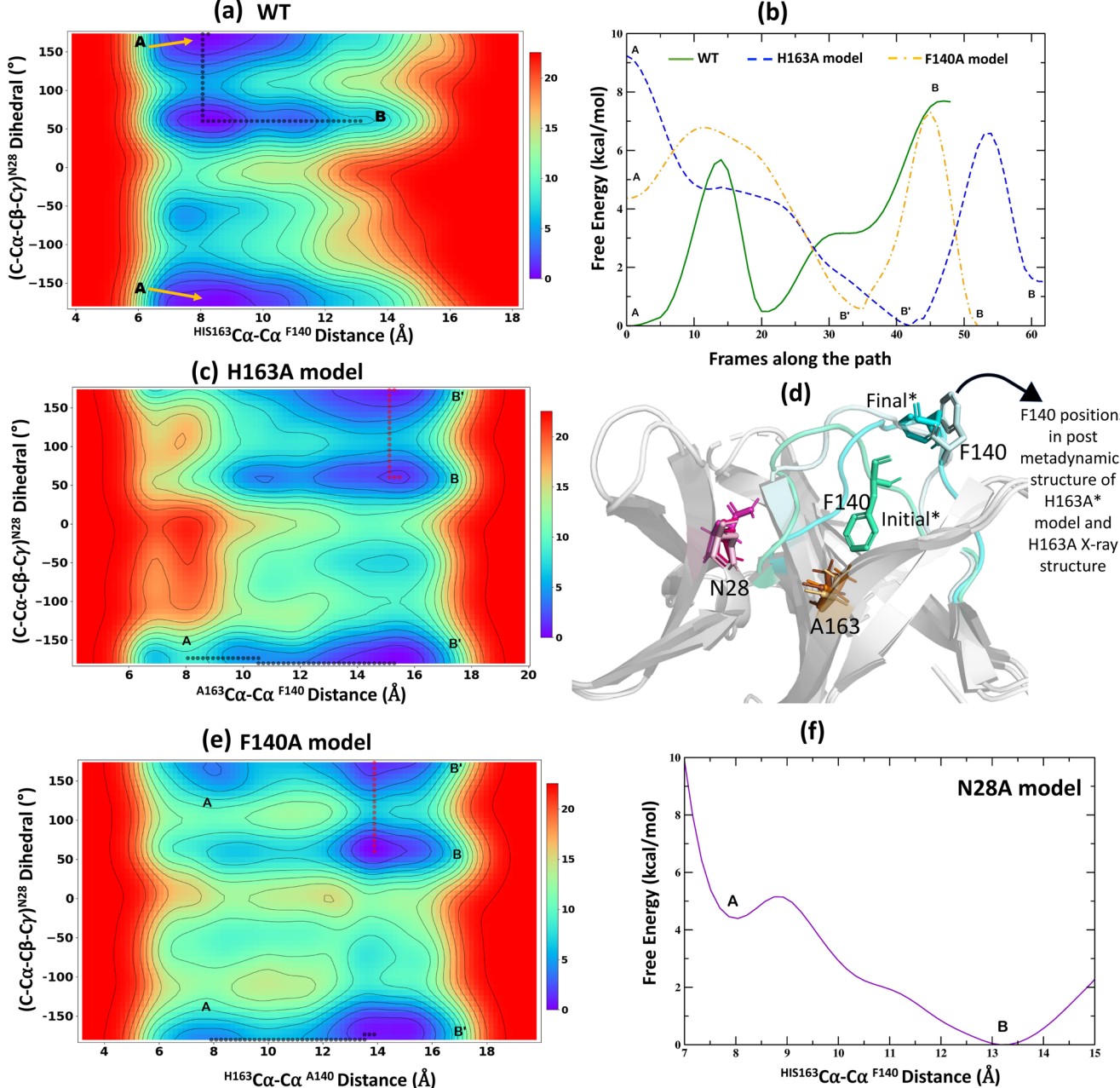

**Fig. 6 | 2D free-energy surfaces and 1D profiles for wildtype and mutant models using well-tempered metadynamics simulations.** Free-energy surfaces describing the transitions from state A (WT conformation) to state B (conformation closer to H163A crystal structure) in the WT (**a**), H163A (**c**), and F140A model (**e**) are shown. The surfaces were explored for the changes in two CVs, the distance between Cα atoms of amino acids at 140 and 163 positions and the side chain dihedral rotation of N28. The resultant surfaces are depicted in a red to blue spectrum that corresponds to high- and low-energy structures, respectively, and the states along the path are marked. The comparison of the 1D free-energy profiles corresponding to the minimum-energy paths connecting states A and B in the WT, H163A, and F140A models are shown in (**b**). The superimposed structures of the H163A crystal structure against the initial and the final states from the metadynamics simulation of the H163A model are described in (**d**). In the initial state of the H163A model, the side chain of F140 (green cyan) was present in the inward conformation, whereas, at the end of the simulation, the active site loop was dislocated, and the F140 side chain was exposed to the surface—a conformation similar (light cyan) to that seen in the mutant crystal structure. The side chain rotation of N28 (purple) is also shown. **f** Free-energy profile for the N28A model shows a free fall from state A to state B due to a lack of any significant barrier in its path.

significant barrier associated with breaking of the H163-F140 aromatic stacking (Supplementary Fig. 12). Therefore, our simulation highlights that breaking the association of N28 with the two cysteine residues is a key process (with a high free-energy barrier) in facilitating the C117-C145 disulfide bond. This is also supported by the earlier disulfide-bonded structure of the N28A SARS-CoV-1 Mpro[28]. This work demonstrates that the rotation of N28 can be achieved by breaking the H163-F140 interaction through a H163A point mutation.

Mpro contains an unusually large number of histidine residues, and their protonation states impact enzyme catalysis, substrate binding, and ligand interactions in Mpro[45,46]. In particular, the neutral state (epsilon protonated) of H163 was shown as critical for maintaining a stable hydrogen bond with the substrate and the inhibitors. A previous MD-based study suggested that protonation of H163 decreased the volume of the lateral binding pocket and subsequently reduced the ligand binding affinity to Mpro[43]. Complementing these insights, our

current work provides experimental evidence demonstrating the importance of H163 in regulating the active conformation of SARS-CoV-2 Mpro. Overall, our X-ray crystal structures and metadynamics simulations unravel a link between the H163-F140 aromatic stacking at the lateral pocket of Mpro and the catalytic residue C145. Disturbing residues in this intricate molecular network could alter Mpro's underlying conformational free-energy landscape, reshape the active site, and trigger the alternate inactive conformation.

## Summary of findings

Given the importance of Mpro in the replication of coronaviruses, significant efforts have gone into understanding its structure–function relationships to lay the foundation for COVID-19 therapeutic development. However, there is still a knowledge gap in the structure–function relationships of redox-associated conformational changes, a phenomenon that may play an important biological role in protecting the enzyme against acute immune system-triggered oxidation. Accumulating scientific evidence suggests that the catalytic cysteine (C145) forms a disulfide bond with the nearby C117 to form an inactive, oxidized conformation. An earlier crystal structure of SARS-CoV-1 Mpro showed that this is possible when N28, a residue that bridges the two cysteines, is mutated to an alanine[28]. In this work, we demonstrate that a similar disulfide-bonded conformation in SARS-CoV-2 Mpro can be achieved by making a point mutant (H163A) in a lateral pocket in the enzyme. Our crystal structures and metadynamics simulations provide a working hypothesis through which the H163A mutation reduces the free-energy gap between the reduced and oxidized conformations, facilitating disulfide bond formation (details are summarized in Supplementary Discussion). Our work suggests that both WT and H163A enzymes explore both active and inactive states, with the mutant enzyme favoring of the inactive form at equilibrium. Thus, our H163A structure potentially presents a platform for developing small-molecule inhibitors that function by either triggering or locking this oxidized, inactive conformational state in the SARS-CoV-2 Mpro and possibly other homologous coronavirus proteases. Despite the mechanistic link between the C117-C145 disulfide bond and the NOS bridge remaining unclear, our structure provides a tool for further uncovering redox-associated structure–function relationships in Mpro.

## Methods

### General information

Protein purification supplies were purchased from ChemImpex Inc. (Wood Dale, IL, USA) with the exception of IgePal-CA630, which was purchased from Thermo Fisher Scientific (Waltham, MA, USA). Crystallization screens were purchased from Molecular Dimensions (Holland, OH, USA). Crystallization solutions were reproduced with chemicals purchased from ChemImpex Inc. (Wood Dale, IL, USA). Crystal mounts, loops, and a UniPuck system were purchased from MiTeGen (Ithaca, NY, USA).

### Cloning and mutagenesis of Mpro expression vector

The SARS-CoV-2 WT Mpro gene (UniProtKB P0DTD1; nsp5 sequence) was codon optimized for *E. coli* and cloned into a SUMOstar vector (LifeSensors Inc.; Malvern, PA, USA) as a C-terminal fusion to an N-terminal small ubiquitin-related modifier (SUMO) tag using *BsaI* and *XhoI* (GenScript; Piscataway, NJ, USA). This results in a construct that, upon SUMO cleavage, generates a protein with the native N-terminus. The H163A mutant was generated in the background of this WT construct (GenScript).

### Protein expression and purification

The protein expression and purification protocols for both WT and H163A Mpro were identical. Heterologous expression of Mpro was done in *Escherichia coli* strain BL21(DE3). An overnight culture was inoculated into ZYP-5052 autoinduction media[47] at a ratio of 50 mL overnight culture to 1 L final media volume with a minimum headspace to media ratio of 1:1. ZYP-5052 media was supplemented with 50 μg/mL kanamycin and cells were grown at 20 °C at 150 rpm for 40–48 h, harvested at 6000×$g$, and cell pellets were stored at −80 °C.

All purification steps were carried out at 4 °C. Cell pellets were thawed in Buffer A (25 mM HEPES, pH 7.5, 0.5 M NaCl, 10 mM imidazole, 1 mM TCEP), passed twice through a FRENCH Pressure Cell (Thermo Fisher Scientific; Waltham, MA, USA) at 1100 psi for cell lysis, and cell debris was removed via high-speed centrifugation at 17,000×$g$. The clarified cell lysate was then incubated with NiNTA resin (Qiagen; Hilden, Germany) and pre-equilibrated in Buffer A, for 1 h. The resin was first washed with 10 column volumes (CVs) of Buffer B (25 mM HEPES, pH 7.5, 0.1% (v/v) IgePal CA630, 10 mM imidazole, 1 mM TCEP) to remove non-specific hydrophobically bound contaminants, followed by a wash with 15 CVs of Buffer A. The protein was eluted with Buffer C (25 mM HEPES, pH 7.5, 0.5 M NaCl, 300 mM imidazole, and 1 mM TCEP) and digested with SUMO protease overnight (expressed and purified in-house from Addgene plasmid pCDB302[48]). The cleaved protein was dialyzed against Buffer D (25 mM HEPES, pH 7.5, 0.5 M NaCl, and 1 mM TCEP) twice overnight to remove residual imidazole. The protein was then incubated with a second round of NiNTA resin, equilibrated in Buffer D, to remove the cleaved SUMO tag and any remaining uncleaved Mpro fusion. NiNTA flow-through was concentrated to less than 1 mL and loaded onto a pre-packed HiLoad Superdex 75 pg 16/600 (Cytiva; Marlborough, MA, USA), pre-equilibrated in Crystallization Buffer (20 mM Tris, pH 8.0, 150 mM NaCl, and 1 mM TCEP), and run at 0.5 mL/min. The purity of the protein in the non-aggregate absorbance peak was qualitatively analysed using SDS-PAGE. Pure fractions were concentrated, frozen in pellets at 80 mg/mL for WT Mpro and 50 mg/mL for H163A Mpro by direct immersion in liquid nitrogen, and stored at −80 °C. Protein concentration was measured using a 1% mass extinction coefficient of 9.73 M$^{-1}$ cm$^{-1}$, theoretically determined by Mpro's primary sequence[49].

### Differential scanning calorimetry (DSC)

WT and H163A Mpro was first dialyzed overnight out of Crystallization Buffer into DSC Buffer (25 mM HEPES, pH 7.5, and 1 mM TCEP) using small-volume dialysis tubes (D-tube Dialyzer Mini; Millipore Sigma, Burlington MA, USA). Protein concentration was re-measured using the same 1% mass extinction coefficient as described previously. About 350 μL of DSC Buffer previously used for dialysis was filtered, degassed, and loaded into the reference and sample cells of a MicroCal VP-DSC MicroCalorimeter (Northampton, MA, USA) and left to equilibrate overnight while the instrument began scanning from 15 to 65 °C at 60 °C/h. The instrument was equilibrated and data were collected under pressure of ~40 psi. About 350 μL of 0.5 mg/mL (14.8 μM) WT or H163A Mpro was injected into the sample cell when the instrument reached 30 °C on a downscan to thermally equilibrate the sample prior to collecting the experimental thermogram.

### Fluorescent kinetic assay

WT Mpro activity was assayed in triplicate at 25 °C using a Tecan Infinite M1000 plate reader (360 nm excitation, 490 nm emission, 5 nm bandwidths), collecting data every 20 s for 10 min. The fluorescent substrate, DABCYL-KTSAVLQSGFRKM-E(EDANS) (GenScript), was solubilized in 100% DMSO and aliquoted for storage at −80 °C, then thawed as needed to create substrate stocks. Protease activity was assayed in 20 mM Tris pH 8.0, 150 mM NaCl, 1 mM TCEP, and 3% (v/v) DMSO, using 0.5 μg of enzyme (150 nM) per 100 μL reaction volume. To enzymatically characterize the protease, final concentrations of the substrate were varied between 0 and 90 μM (higher concentrations were not soluble in 3% (v/v) DMSO). Initial rates of reaction were taken from the slopes calculated from a simple linear regression of the first 5 min of data. Raw data in relative fluorescent units were converted to

molarity using a standard curve with a fully cleaved substrate. The Michaelis-Menten curve for WT Mpro was plotted as $s^{-1}$ vs substrate concentration, and fit to the Hill equation[50]. H163A Mpro activity was assayed identically to the WT Mpro. No activity was seen using 150 nM enzyme, even when the reaction was supplemented with 10 mM β-mercaptoethanol, DTT, or TCEP. Detectable changes in signal were seen when the enzyme was assayed at 100 µM against a maximum concentration of substrate (90 µM).

## Protein crystallization

To obtain crystals of the H163A Mpro in complex with GC376, 5 mg/mL H163A mutant (~148 µM) was incubated with 400 µM GC376 (BPS Bioscience; San Diego, CA, USA) at room temperature for 2 h prior to setting up the crystallization experiment. Thin plate-like crystals in a "flower" arrangement appeared after several days from drops mixed from 2.0 µL protein sample and 2.0 µL reservoir solution (0.1 M Bis-Tris, pH 6.5 and 25% (w/v) PEG 3350) supplemented with 3% (v/v) DMSO. These crystal clusters were manually manipulated to acquire single crystals suitable for diffraction. High-throughput crystallization trials for the H163A mutant were carried out with commercially available screens in small-volume sitting-drop trays using a Crystal Gryphon LCP robot (Art Robbins Instruments; Sunnyvale, CA, USA). Drops consisting of 0.2 µL protein sample (20 mg/mL H163A Mpro in Crystallization Buffer) mixed with 0.2 µL reservoir solution were left to equilibrate at room temperature for a few weeks. Initial hits grew after one week in 0.1 M Tris, pH 8.5, and 22% (v/v) PEG Smear Broad (BCS B11)[51]. The optimized condition yielded crystals in stacked plates from a mix of 2.0 µL protein sample (20 mg/mL in Crystallization Buffer) and 2.0 µL reservoir solution (0.1 M Tris, pH 8.5 and 26% (v/v) PEG Smear Broad). To obtain reduced crystals of H163A Mpro, the above crystals were soaked with the same reservoir solution supplemented with 20 mM TCEP for two hours at room temperature prior to being plunge frozen in liquid nitrogen. All crystals were cryoprotected in their respective reservoir solutions supplemented with 20% (v/v) glycerol.

## X-ray data collection and structure refinement

Diffraction data were collected at the Cornell High-Energy Synchrotron Source (CHESS) ID7B2 beamline on a Detectris Pilatus3 S 6 M. Data were indexed, integrated, and scaled with DIALS (version 3.8.0)[52] and imported into CCP4i suite (version 8.0.009)[53] with AIMLESS (version 0.7.9)[54]. Molecular replacement (MOLREP version 11.9.02)[55] for all structures were done with a high-resolution WT Mpro model (PDB 7ALH[56]). Refinement was done using phenix.refine (version 1.20.1_4487)[57] in conjunction with manual model building in COOT (version 0.8.9.2)[58]. Translation-libration-screw parameters were automatically determined and used by phenix.refine for all structures. Non-crystallographic symmetry restraints were kept throughout refinement for only the reduced H163A Mpro structure because of the lower resolution. Due to the crystallographic evidence for alternate conformations of G138-V148 in the apo H163A structure, especially in chain B, the occupancies of these residues were automatically refined with phenix.refine due to the difficulties in accounting for the alternate conformations in the model. Model geometry was analysed and optimized based on suggestions by MolProbity (version 4.5.2)[59]. Data collection and model statistics are summarized in Supplementary Table 1.

## Small-angle X-ray scattering

SAXS data were collected at the Cornell High-Energy Synchrotron Source (CHESS) ID7A beamline using an X-ray wavelength of 1.1013 Å and an EIGER 4 M detector[60]. H163A Mpro was dialyzed against 25 mM HEPES, pH 7.5 with 1 mM TCEP using a 12–14 kDa MWCO D-Tube Mini Dialyzer (Millipore Sigma; Burlington MA, USA) to obtain a matching solution for buffer subtraction. About 30 µL of dialyzed buffer and a variety of concentrations ranging from 0.25 to 6.3 mg/mL H163A Mpro, diluted with the matched buffer, were injected into the sample

capillary for analysis. Samples were oscillated within the capillary during data collection to reduce radiation damage. Fifteen images were collected for each sample, with the detector being exposed for one second per image with no attenuation. Images were azimuthally averaged and buffer subtracted in RAW (version 2.1.4)[61]. Processed data were subsequently analysed in RAW[61], GNOM (ATSAS package, version 3.2.1)[62], and FoXS (web server, accessed April 17th, 2023)[63].

## Modeling and metadynamics simulations

Metadynamics simulations were performed for the WT (PDB 7JUN)[64], H163A, F140A, and N28A model systems. All simulations were performed on the monomer form to understand the free-energy landscape associated with the exposure of F140 to the surface, the perturbation of the active site loop, and breaking of the C145-N28-C117 network. All the mutant models were constructed by performing a single point mutation to the WT structure using the Chimera-alpha program[35]. Each of the four systems were solvated in a 97 Å wide cubic box of TIP3P water molecules[65], and electro-neutralized to a physiological ionic concentration of 0.15 mM using sodium and chloride ions. The modeled systems were energy minimized in 1000 steps using the steepest descent algorithm[66] and subsequently subjected to a multi-stage equilibration under the conditions of a constant temperature of 310 K and constant pressure of 1 bar, which were maintained using Berendsen thermostat and barostat[67]. Initially, the systems underwent 500 ps of equilibration with a harmonic restraint of 25 kcal/mol/Å² on the heavy atoms of the protein. The restraints were gradually reduced from 25 to 0.78 kcal/mol/Å² in six consecutive steps of 50 ps long equilibration (by reducing the restraints by half in each step as 25→12.5→6.25→3.125→1.56→0.78 kcal/mol/Å²). Finally, the systems underwent 1 ns unrestrained equilibration with constant temperature 310 K and 1 bar pressure using V-rescale thermostat[68] with a coupling constant of 1 ps and Parrinello–Rahman barostat[69] with a coupling constant of 1 ps. Throughout all stages of equilibration, the electrostatic interaction were treated using Particle mesh Ewald (PME)[70] with a cut-off of 10 Å and van der Waals cut-off was taken to be 10 Å. All MD simulations were carried out using the GROMACS-2019.6[71] program and AMBERff14SB forcefield[72] for describing the model systems. The initial and final coordinates of the WT and mutant models from classical MD simulation are provided in Supplementary Data 1 (WT), 2 (H163A), 3 (F140A), and 4 (N28A).

The equilibrated systems were used as the starting structures for performing well-tempered metadynamics simulations using the PLUMED-2.6.2[73] patch available for the GROMACS-2019.6 program. In well-tempered metadynamics simulations, we employed two sets of collective variables (CVs) to accelerate the transition of the WT conformational state A to the alternate state B that is closer to that seen in the H163A crystal structure. The two CVs include (1) the distance between the Cα atoms of the amino acids at position 163 and 140 (sampled between 8–16 Å); and (2) the side chain dihedral rotation of N28 (sampled between −180° to +180°), which are the most prominent changes amongst our WT and H163A crystal structures. During metadynamics, a history-dependent Gaussian bias was applied at a regular interval of 1 ps to fill the free-energy wells and to efficiently explore the free-energy landscape involving the conformational transition from state "A" to "B". We used the initial Gaussian height of 1 kJ/mol and the Gaussian width of 0.5 Å for the distance CV, along with 0.35 radians for the dihedral CV. We used a bias factor of 5 to reduce the Gaussian height so as to prevent the over-filling of the free-energy wells during metadynamics simulations. Metadynamics simulations for each of the model systems were run for 150 ns. The minimum free-energy paths for transitioning from state "A" to "B" were computed using different lengths of MD trajectories (50, 75, 100, 125, and 150 ns) to confirm the convergence of the profiles. We observed that the free-energy profiles were converged for 125 and 150 ns trajectories (see Supplementary Figs. 6–9). The minimum

free-energy paths corresponding to state transitions were computed using the MEPSA-v1.4 software[74]. All structure visualization and analyses of simulation trajectories were performed using VMD-1.9.3 software[75] and the plots were generated using Grace-5.1.25[76] plotting tool and MEPSA-v1.4 software. The initial and final coordinates of the WT and mutant models from metadynamics simulations are provided in Supplementary Data 5–8, and the coordinates of the conformations sampled along their free-energy paths from metadynamics are provided in Supplementary Data 9–12.

### Reporting summary

Further information on research design is available in the Nature Portfolio Reporting Summary linked to this article.

## Data availability

Raw experimental data and metadynamics trajectory files that are needed to recapitulate the results of this paper can be requested from the authors. The WT Mpro structure used for all the modeling and simulation in this work are available with the PDB accession code 7JUN. The initial and final coordinates of the WT and mutant models from classical MD simulations in this work are provided in Supplementary Data 1–4. The initial and final coordinates of the WT and mutant models from metadynamics simulations in this work are provided in Supplementary Data 5–8. The conformations sampled along the free-energy paths of the WT and mutant Mpro models are described in Supplementary Figs. 9–12 are provided as single aligned PDB files for each system in Supplementary Data 9–12. The previously published crystal structure of WT SARS-CoV-2 Mpro used in this work for molecular replacement is available with the PDB accession code 7ALH. Maps and models for a variety of Mpro structures that were referenced in the text and figures but were not directly used to process data can be found in the PDB with the following accession codes: 6XB0, 6XMK, 7JR4, 3FZD, 7TGR, and 7BB2. The maps and models for the GC376-bound H163A Mpro, apo H163A Mpro, and reduced H163A Mpro structures are available in the PDB accession codes, 8DD6, 8DDL, and 8SG6, respectively. SAXS data for the 0.25, 0.5, 1.0, 3.0, and 6.3 mg/mL H163A Mpro have been deposited in the SASBDB with database IDs of SASDSP5, SASDSQ5, SASDSR5, SASDSS5, and SASDST5, respectively. Source data are provided with this paper.

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

## Acknowledgements

This work was supported by seed funding from the Centre for Bioengineering and Biotechnology (A.G. and S.K.), the Science Action Response Fund (S.K. and T.H.) at the University of Waterloo, Faculty of Science, along with the Natural Science and Engineering Research Council (NSERC) of Canada (T.H. and N.T.). This research was undertaken thanks in part to funding from the Canada First Research Excellence Fund (A.G. and S.K.). The modeling and simulation presented in this work was enabled by the support provided by the Digital Research Alliance of Canada through the Resource Allocation Competition 2022 (A.G. and S.K.). We would like to thank the beamline scientists at CHESS (ID7B2 for crystallography and ID7A for SAXS) for their assistance and for providing us with beam time. CHESS is supported by the NSF & NIH/NIGMS via NSF award DMR-1829070, and the MacCHESS resource is supported by NIH/NIGMS award GM-124166. We would also like to thank Julia Solonenka for her help with the initial optimization of the Mpro fluorescent kinetic assay and Harmeen Deol at Dr. Elizabeth Meiering's Lab at the University of Waterloo for training and access to the DSC instrument.

## Author contributions

S.K., T.H., and A.G. conceived of the project. N.T., S.K., T.H., and A.G. designed the experiments. N.T. expressed and purified WT and H163A Mpro. N.T., M.J.M., and S.A.E.B. crystallized and collected crystallographic data on the H163A Mpro crystals. N.T. and M.J.M. processed and refined the crystallographic data. N.T. collected the DSC data. N.T. and M.J.M. collected and analyzed the SAXS data. S.A.E.B. optimized and kinetically characterized both WT and H163A Mpro using the fluorescent kinetic assay. S.D. performed all metadynamics simulations and analyzed the results with the help of S.K. and A.G. N.T. wrote the introduction and non-metadynamics results/discussion sections and generated all experimental data figures. A.G. and S.D. drafted the discussions and generated figures related to metadynamics simulations. All authors edited the manuscript.

## Competing interests

The authors declare the following competing interests: a patent application on the inactive, disulfide-bonded H163A Mpro structure and the discovery that H163 with alanine leading to this unique conformation have been filed in the United States Patent and Trademark Office (serial number: 18/332.095). A.G., T.H., S.K., and N.T. are co-inventors of this patent. The remaining authors declare no competing interests.
