## [Peer Review File · Nature Communications]

REVIEWER COMMENTS

Reviewer #1 (Remarks to the Author):

The authors report on post-translational modifications of the SARS-CoV-2 main protease (Mpro) that is key drug target for fighting the current pandemic. Specifically, they observe a structural switch in an active site variant (H163A) that leads to formation of a disulfide between the catalytic cysteine C145 and C117. This redox switch had been postulated but experimental structural data were missing so far. Also, they observe a covalent crosslink between a surface lysine and a neighboring cysteine recently proposed to be part of a NOS/SONOS switch. The authors explore the mechanism of switching by studying further variants that are part of the hinge and by metadynamics analysis.

This reviewer considers this work as being important as a novel conformation of Mpro is reported that offers new opportunities in drug design and can be a springboard for novel classes of inhibitors. Several minor questions remain.

1. The authors report that even with relatively high concentrations of reductants no further increase in activity could be increased. How long did they react the oxidized protein with reductants? It may require a couple of hours to reestablish enzymatic activity.

2. Did the authors obtain experimental evidence that the monomer-dimer equilibrium was affected in the tested Mpro variants?

Reviewer #2 (Remarks to the Author):

The manuscript is a well-written report on the crystal structure of an interesting mutant of Mpro , H163A, together with some simulations investigating the associated energetics. The interest is that this structure shows an oxidized conformation with the catalytic cysteine in a disulfide bond. The following points need to be considered:

(1) The relevance of H163A to physiological conditions i.e. a druggable state appears to be somewhat tenuous. For example an extra water molecule is found in the active site. The idea that Mpro adopts this conformation under oxidative stress is a hypothesis for which little direct evidence is presented. However, S-S bonds are indeed detectable experimentally using non-crystallographic methods. The authors should consider using these experiments to test for WT Mpro S-S bond formation under oxidative stress.

(2) The crystal structure in Ref 18 is already oxidized without the need for mutation. Can the authors comment on this finding?

(3) Both protonation states and inhibitor binding can greatly influence the type of equilibria discussed here. The authors should discuss their findings on redox variability in comparison to the discussions on protonation states and inhibitor binding reported in *Chem. Sci.*, 2021, 12, 1513–1527 and *J. Phys. Chem. Lett.* 2021, 12, 4195–4202 with a view to establishing a putative grand scheme of Pro conformational states.

REVIEWER COMMENTS

Reviewer #1 (Remarks to the Author):

General comment: The authors report on post-translational modifications of the SARS-CoV-2 main protease (Mpro) that is key drug target for fighting the current pandemic. Specifically, they observe a structural switch in an active site variant (H163A) that leads to formation of a disulfide between the catalytic cysteine C145 and C117. This redox switch had been postulated but experimental structural data were missing so far. Also, they observe a covalent crosslink between a surface lysine and a neighboring cysteine recently proposed to be part of a NOS/SONOS switch. The authors explore the mechanism of switching by studying further variants that are part of the hinge and by metadynamics analysis.

This reviewer considers this work as being important as a novel conformation of Mpro is reported that offers new opportunities in drug design and can be a springboard for novel classes of inhibitors. Several minor questions remain.

Response: We thank the reviewer for their positive feedback on our manuscript and for raising some important questions, which helped us gather additional experimental data to support our findings. Our point-by-point responses to the reviewer's questions are provided below.

1. The authors report that even with relatively high concentrations of reductants no further increase in activity could be increased. How long did they react the oxidized protein with reductants? It may require a couple of hours to reestablish enzymatic activity.

Response: The H163A mutant was incubated with 10 mM TCEP for one hour prior to performing the kinetic assay. This information was added to the manuscript (lines 181-183 numbered according to the "simple markup" document).

Although there may be some time-dependent effect following reduction to reestablish activity, we have added an additional crystal structure to our manuscript to suggest that the inactive conformation, whereby the F140 residue is in the "out" position, is maintained even when the disulfide bond between C117 and C145 is reduced after the mutant crystal is soaked with 20mM TCEP for 2 hours. This suggests that the mutant would still be much less active than the WT enzyme even if the disulfide bond was fully reduced in solution. This information is detailed in lines 296-303 of the manuscript. We have also added a new figure (*Supplementary Fig. 8*) in the supplementary information.

This new discussion (on Page 10) and *Supplementary Fig. 8* in the revised manuscript are copied below:

When the mutant crystals were soaked with 20 mM TCEP for two hours, the resultant crystal structure (*Supplementary Table 1*) was fully reduced as both the NOS bridge and the C117-C145 disulfide bonds were broken in both protomers (*Supplementary Fig. 8a*).

Interestingly, despite the breakage of the disulfide bond, the active-site loop was not repositioned as the F140 side chain remained flipped in the “out” position (*Supplementary Fig. 8b*). This suggests that the disulfide bond stabilizes this inactive state but is not sufficient for generating this conformation. The disulfide bond is thus a consequence of placing the two cysteine residues in proximity following displacement of the active-site loop. Our work therefore indicates that the loss of the π -stacking interaction between F140 and H163 is the source of the inactive conformation of Mpro.

Supplementary Figure 8 – Reduced Structure of the H163A Mpro Mutant. (a) Soaking the H163A Mpro crystals with 20mM TCEP reduces the C117-C145 disulfide bond, which returns the active-site loop to a more WT-like state. (b) Despite reduction of this disulfide bond, the enzyme remains in an inactive state due to the F140 side chain being in the “out” position, with a similar conformation to the disulfide-bonded H163A structure depicted in *Fig. 5a* in the main text. The WT structure (PDB 7CAM) is shown in grey for comparison. Interestingly, the N-terminus of the reduced H163A structure is in a WT-like conformation whereas the disulfide-bonded H163A structure has its N-terminus rotated approximately 90° from the WT conformation (*Fig. 4cd* in the main text).

2. Did the authors obtain experimental evidence that the monomer-dimer equilibrium was affected in the tested Mpro variants?

Response: Thanks for this constructive suggestion. We have added additional small-angle X-ray scattering (SAXS) data to our manuscript to show that the H163A mutant is dimeric at a wide range of concentrations, ranging from 0.25 to 6.30 mg/mL. The SAXS profiles collected are very similar to previously published SAXS data on the dimeric WT enzyme (ref. 37). This is corroborated with our DSC and unpublished gel-filtration data. As such, data collected at the various enzyme concentrations tested suggests that the monomer-dimer equilibrium is not significantly affected by the H163A mutation. This information is detailed in lines 202-209 of the manuscript.

This new discussion (on Page 7) and *Supplementary Fig. 4* and *5* in the revised manuscript are provided below:

To further verify the oligomeric state of the H163A mutant, small-angle X-ray scattering (SAXS) experiments were performed at concentrations ranging from 0.25 to 6.30 mg/mL. These SAXS profiles (*Supplementary Fig. 4*) show that the mutant is indeed dimeric, consistent with a previously published WT Mpro SAXS dataset (SASBDB Entry SASDJG5),³⁷ and was notably different from the monomer generated by removing one protomer of the known dimer structure (*Supplementary Fig. 5*). This suggests that the monomer-dimer equilibrium is not significantly affected by the H163A mutation at the concentrations of enzyme used in the DSC and SAXS experiments.

Supplementary Figure 4 – Comparing Small-Angle X-ray Scattering Profiles Between the H163A and WT Mpro. The scattering profiles of the H163A mutant are consistent in all regards to previously published dimeric WT Mpro SAXS data (SASBDB Entry SASDJG5).³⁷ The raw scattering curves (*top left*), normalized Kratky (*bottom left*), and normalized $P(r)$ plots (*bottom right*) for various concentrations of H163A Mpro have the same general shape to WT Mpro, showing that the mutant scattering is consistent with a dimer for these concentrations. The good data quality of the representative 3.0 mg/mL data are exemplified by the linear Guinier region (*top of top right plot*) and the fit's non-skewed residuals (*bottom of top right plot*), which was fit to a $q_{\max}R_g$ of ~ 1.32 . Data were processed with *RAW*⁵⁸ and *GNOM*.⁵⁹ Raw scattering curves were offset in $I(q)$ by factors of 50 to aid with visual comparison. The error bars in the raw scattering curves represent standard error. Data points for the 0.25 and 0.5 mg/mL normalized $P(r)$ plots are depicted as smaller points to reduce visual clutter due to the noisier data.

Supplementary Figure 5 – Fitting Simulated Scatter of Crystal Structures to H163A Mpro SAXS Data. Simulated scattering profiles generated from crystal structures of the apo H163A mutant dimer (black fit; PDB 8DDL; the dimer is the asymmetric unit) and a WT monomer model (red fit; PDB 7ALH; one protomer of the dimer is in the asymmetric unit) were fit to the experimental scattering profile of H163A Mpro at 3.0 mg/mL using the FoXS webserver.⁶⁰ The simulated scatter from the Mpro dimer reasonably fits the experimental data ($\chi^2 = 2.1$) while the simulated scatter from the monomer model does not fit the data ($\chi^2 = 56.6$).

Reviewer #2 (Remarks to the Author):

The manuscript is a well-written report on the crystal structure of an interesting mutant of Mpro , H163A, together with some simulations investigating the associated energetics. The interest is that this structure shows an oxidized conformation with the catalytic cysteine in a disulfide bond. The following points need to be considered:

Response: We thank the reviewer for their positive views on our manuscript and for providing constructive suggestions for revision.

(1) The relevance of H163A to physiological conditions i.e. a druggable state appears to be somewhat tenuous. For example an extra water molecule is found in the active site. The idea that Mpro adopts this conformation under oxidative stress is a hypothesis for which little direct evidence is presented. However, S-S bonds are indeed detectable experimentally using non-crystallographic methods. The authors should consider using these experiments to test for WT Mpro S-S bond formation under oxidative stress.

Response: We understand the reviewer's point regarding the druggable state of this new conformation, as this is not the focus of this work. We have therefore revised our title by removing the aspect related to therapeutic design.

Funk et al. (2022) (ref. 20) details non-crystallographic, mass spectrometric experimental data supporting the formation of the C117-C145 disulfide bond, among other structural oxidation effects such as the SONOS bridge, under conditions that mimic oxidative stress. This is discussed on Page 4 (lines 111-117 numbered according to the "simple markup" document) within our introduction section. It is important to note that these structural features are present in our H163A mutant structure without adding exogenous oxidizing agents.

(2) The crystal structure in Ref 18 is already oxidized without the need for mutation. Can the authors comment on this finding?

Response: *Kneller et al.* (2020) (ref. 18) describes an oxidized Mpro structure whereby C145 is oxidized as a peroxysulfenate. These experiments involved transferring the crystal out of reducing agent for an hour, thereby purposefully oxidizing the enzyme, to observe structural changes resulting from oxidization. Our H163A mutant structure shows an oxidized state despite being purified and crystallized with reducing agent and without any additional conditions for purposeful oxidation.

Our findings, therefore, not only describe an oxidized form of Mpro but also unravel an important link between H163 (in the lateral pocket) and its enzymatic state.

(3) Both protonation states and inhibitor binding can greatly influence the type of equilibria discussed here. The authors should discuss their findings on redox variability in comparison to the discussions on protonation states and inhibitor binding reported in Chem. Sci., 2021, 12, 1513–1527 1513 and J. Phys. Chem. Lett. 2021, 12, 4195–4202 with a view to establishing a putative grand scheme of Pro conformational states.

Response: Thanks for bringing these interesting research papers to our attention. These papers highlight the importance of the protonation states of histidine residues, particularly H163, towards substrate and ligand binding affinity. Our work expands these insights further to show that H163 renders a stacking interaction with F140, which in turn is critical for retaining the shape of the binding pocket and, hence, the active state of Mpro. This discussion is added on Page 13 and 14 (lines 400-411) in our revised manuscript, which is also given below:

Mpro encompasses an unusually large number of histidine residues, and their protonation states impact enzyme catalysis, substrate binding, and ligand interactions in Mpro.^{43,44} In particular, the neutral state (epsilon protonated) of H163 was shown as critical for maintaining a stable hydrogen bond with the substrate and the inhibitors. A previous MD-based study suggested that protonation of H163 decreased the volume of the lateral binding pocket and subsequently reduced the ligand binding affinity to Mpro.⁴³ Complementing these insights, our current work provides experimental evidence demonstrating the importance of H163 in regulating the active conformation of SARS-CoV-2 Mpro. Overall, our X-ray crystal structures and metadynamics simulations unravel a link between the H163-F140 aromatic stacking at the lateral pocket of Mpro and the catalytic residue C145. Disturbing residues in this intricate molecular network could alter Mpro's underlying conformational free-energy landscape, reshape the active site, and trigger the alternate inactive conformation.